# Robustness of Explainable Artificial Intelligence in Industrial Process Modelling

**Benedikt Kantz** [1] **Clemens Staudinger** [2] **Christoph Feilmayr** [2] **Johannes Wachlmayr** [3] **Alexander Haberl** [2] **Stefan Schuster** [2] **Franz Pernkopf** [1]

## Abstract

eXplainable Artificial Intelligence (XAI) aims at providing understandable explanations of black box models. In this paper, we evaluate current XAI methods by scoring them based on ground truth simulations and sensitivity analysis. To this end, we used an Electric Arc Furnace (EAF) model to better understand the limits and robustness characteristics of XAI methods such as SHapley Additive exPlanations (SHAP), Local Interpretable Model-agnostic Explanations (LIME), as well as Averaged Local Effects (ALE) or Smooth Gradients (SG) in a highly topical setting. These XAI methods were applied to various types of black-box models and then scored based on their correctness compared to the ground-truth sensitivity of the data-generating processes using a novel scoring evaluation methodology over a range of simulated additive noise. The resulting evaluation shows that the capability of the Machine Learning (ML) models to capture the process accurately is, indeed, coupled with the correctness of the explainability of the underlying data-generating process. We furthermore show the differences between XAI methods in their ability to correctly predict the true sensitivity of the modeled industrial process.

## 1. Introduction

ML approaches have the power to model complex dependencies in demanding tasks such as industrial processes. However, the behavior of these industrial processes that rely on complex, non-linear interactions is often not fully understood. This results in the need for algorithms to understand and interpret how these ML models arrive at certain predic-

tions and how they might react to certain perturbations in the input. In the last years, there has been an effort to provide explanations to the ML model predictions using XAI (Lundberg & Lee, 2017; Ribeiro et al., 2018; Alvarez-Melis & Jaakkola, 2018; Shrikumar et al., 2017).

Most of these works, even if they focus on the robustness and trustworthiness of the XAI method, have the shortcoming that they can only be evaluated through surrogate measures (Crabbé & van der Schaar, 2023), and the ground truth sensitivity of the evaluated datasets cannot be properly calculated (Alvarez-Melis & Jaakkola, 2018). Some existing approaches rather use data augmentation (Sun et al., 2020) or create measures estimating the importance of the features (Yeh et al., 2019); further related work is provided in Section A.3. None of these systems, to the best of our knowledge, consider the ground truth sensitivity, or gradient, of the data-generating process that created the dataset. Modeling the sensitivity to the inputs is, however, key to understand the underlying process using proxy ML models solely learned on data.

In this paper, we introduce data-driven evaluation of different XAI methods using a simulated process of an EAF model and its ground truth sensitivity, providing insights into the actual limits and robustness properties of state-of-the-art ML models and interpretability approaches. We propose to use a specifically generated dataset and perform perturbations to analyze this robustness empirically. Two central problems, however, arose when scoring these XAI methods and comparing them to a ground truth sensitivity:

1. The feature importance scores over the feature dimensions are not within the same magnitude and range, requiring scaling (Shrikumar et al., 2017).
2. The relative sizes of different XAI feature importance scores are not necessarily aligned to each other (Lundberg & Lee, 2017; Apley & Zhu, 2019), requiring normalization.

Therefore, we introduce a novel evaluation methodology, for solving both of these problems. We essentially analyze how well the XAI methods explain the sensitivity of the input features, compared to a known ground truth sensitivity.

[1] Signal Processing and Speech Communication Laboratory, Technical University Graz, Graz, Austria [2] voestalpine Stahl GmbH, Linz, Austria [3] K1-MET GmbH, Linz, Austria. Correspondence to: Benedikt Kantz <benedikt.kantz@tugraz.at>, Franz Pernkopf <pernkopf@tugraz.at>.

*Accepted at the 1st Machine Learning for Life and Material Sciences Workshop at ICML 2024.* Copyright 2024 by the author(s).

## 2. Data-generating processes

This section outlines the two data generation processes: a toy dataset, as well as the EAF process simulation, including the process variables. These specifically generated datasets are necessary, as other datasets do not provide the ground truth effects $\mathbf{w}_i^*$ of the functions at the datapoints $\mathbf{x}^i$.

### 2.1. Toy dataset

To prove the effectiveness of our evaluation methodology, we first build a small polynomial data-generating system of the form

$$f(x_1, x_2) = k_1 x_1^2 + k_2 x_2^2 + k_3 x_1 x_2 + k_4 x_1 + k_5 x_2 + k_6 \quad (1)$$

with the coefficients $k_d$ being drawn from $k_d \sim \text{Uniform}(0, 1)$ once. The function effects, or gradients, were estimated using automatic gradient calculation using PyTorch (Paszke et al., 2019). This data-generating process was used to generate 1000 samples $f(x_1, x_2)$, where $x_1, x_2 \sim \text{Uniform}(-5, 5)$, which were utilized for the evaluation process.

### 2.2. EAF process

The choice of an EAF model as the data-generating process was owed to a wide range of factors. First, the EAF process itself is relevant in the steel industry as it promises, given enough clean energy, greenhouse gas emission reduction compared to traditional steel production in blast furnaces (De Ras et al., 2019). Furthermore, EAFs have been studied and modeled for a long time using many different approaches and modeling strategies, from their electrical characteristics (Billings et al., 1979; Boulet et al., 2003) to their chemical processes and internal interactions (Zhang & Fruehan, 1995; Basu et al., 2008). These works helped build an accurate chemical simulation of an EAF, representing the real-world system quite well while keeping it simulatable and manageable in parameter space (Binti Ahmad Dzulfakhar et al., 2023). The functional complexity is sophisticated, providing an interesting process to be modeled by ML models. Furthermore, the data generating process is non-independant and identically distrubuted (iid.), as the EAF model has timesteps, and, when considering these, dependencies between each simulated tapping of the furnace arise, making modeling even more challenging. The model itself simulated the individual zones of the EAF reactor as homogeneous zones, with the same temperature and uniform mixture across the whole zone. These zones are the gas zone, solid metal zone, liquid slag zone, liquid metal, and solid slag zone. Additionally, the reactor is modeled as a few discrete parts, particularly the roof, and walls. The liquid metal and slag zones are the ones where measurements were simulated during tapping.

The EAF model (Binti Ahmad Dzulfakhar et al., 2023) was converted into a Python module, where automatic differentiation tools (Paszke et al., 2019) were used to generate ground truth sensitivities of the input parameters. This simulated experiment was repeated over sampled combinations of a subset of different input parameters, which were the oxygen lance rate, oxygen for post-combustion, power of the arc, carbon injection rate, ferromanganese injection rate, and the mass addition rate of solids. Furthermore, auxiliary properties within the simulated tapped material were recorded, more specifically the ratios of silicon dioxide and iron oxide in the slag, as well as the temperature of the liquid slag and metal. The observed target variable, the ratio of carbon in the tapped steel from the furnace, was recorded, too. Each tapping was considered as one data sample $\mathbf{x}_i \in \{\mathbf{x}_1, \ldots, \mathbf{x}_n\}$, consisting of the observed variables. The gradient of the input parameters concerning the output was calculated and accumulated at the timesteps in the simulation, too. This simulation resulted in a dataset of about $n \approx 10^4$ samples after the removal of numerical outliers due to instabilities after simulating the furnace for about a week. The process was furthermore restarted after each parameter change.

### 2.3. Perturbing the Dataset

While the datasets at hand provided a perfect ground truth sensitivity for interpretations, there was still no proper way to assess the robustness of the ML models in combination with the XAI methods. To this end, the dataset was artificially perturbed using noise in two ways. The source of the noise was Gaussian noise added to the feature $j$ of sample $i$, except the target variable, using

$$\tilde{x}_{i,j} = x_{i,j} + n_j \quad (2)$$

where $n_j \sim \mathcal{N}(0, l \cdot (\max_{i \in 1, \ldots, n} x_{i,j} - \min_{i \in 1, \ldots, n} x_{i,j}))$, $x_{i,j}$ being the $j$-th feature of the observation $\mathbf{x}_i = \{x_{i,0}, x_{i,1}, \ldots, x_{i,d}\}$. The 0-th feature is the target $y_i \in \mathcal{Y}$. $l \in [0, 1)$ is the selected noise level of the experiment.

## 3. Scoring Methodology for Local Explanations

We developed an evaluation methodology (see Figure 1) to quantify the effect of the perturbances on the ML models and the interpretation, as there is no such common measure to cater to heterogeneous types of feature importance and interpretability measures, as discussed in Section A.3. Furthermore, the problems that make comparisons difficult, as mentioned in Section 1, were addressed using this scoring methodology.

We denote the output of each XAI method as $\mathbf{w}_i = w_{i,1}, \ldots, w_{i,d}$, independent of the underlying explainer. All ground truth sensitivity values from the data-generating

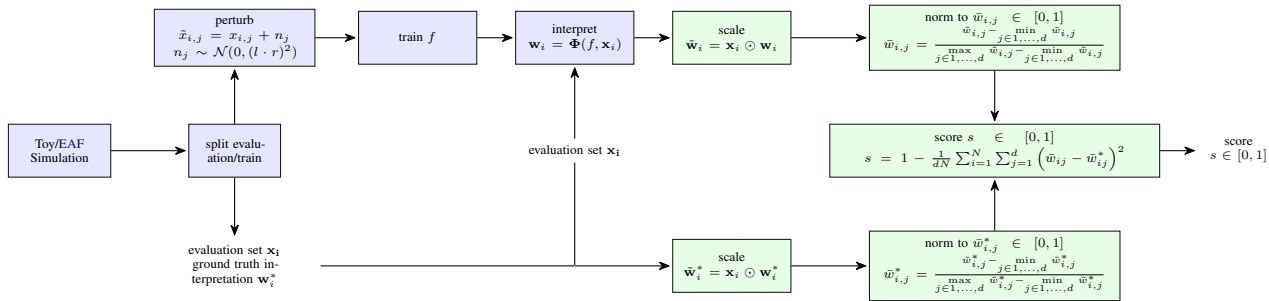

Figure 1: Evaluation methodology, see text for further details and choices of measures.

processes are denoted as $\mathbf{w}_i^*$. All input features of the data-generating functions were used in this evaluation system. To solve the first problem outlined in the introduction, the feature scores $w_{i,j}$ were scaled using the features values themselves, using

$$\tilde{w}_{i,j} = w_{i,j} \cdot x_{i,j}. \tag{3}$$

This helps to adjust the magnitude of the XAI method results to the appropriate scale. The second problem is approached by min-max normalization on each data sample to indicate their relative strength within a sample. The scaling results in an indication of the weights' approximate strength by scaling them to a $[0, 1]$ range using

$$\bar{w}_{i,j} = \frac{\tilde{w}_{i,j} - \min\limits_{j \in 1,...,d} \tilde{w}_{i,j}}{\max\limits_{j \in 1,...,d} \tilde{w}_{i,j} - \min\limits_{j \in 1,...,d} \tilde{w}_{i,j}}. \tag{4}$$

Both scaling operations are performed for the ground truth derivatives $w_{i,j}^*$ of the data-generating process too, leading to the normalized ground truth $\bar{w}_{i,j}^*$. Using both results, the score of the local interpretation can finally be computed using the Brier Score (Brier, 1950), a probabilistic scoring method, initially intended for weather forecasts. It is, in essence, the Mean Squared Error (MSE), and the score $s_i$ for one observation is thus calculated using

$$s_i = \frac{1}{d} \sum_{j=1}^{d} \left( \bar{w}_{i,j} - \bar{w}_{i,j}^* \right)^2. \tag{5}$$

The final score $s$ for one set of $n$ observations is computed by averaging each samples score $s_i$ using

$$s = 1 - \frac{1}{n} \sum_{i=1}^{n} s_i. \tag{6}$$

This averaging should return a score of $s = 1$ if the interpretations completely align with the ground truth and lower if there are discrepancies. This scoring methodology, due to the comparison to the ground truth effect, focuses therefore not on the feature's importance, but rather on the correctness of the effects that a feature, at a certain data point, has.

### 3.1. Evaluation Process

The evaluation methodology is performed 50 times over a randomly sampled fold of 10 percent of the samples from the datasets. However, due to the non-iid data of the EAF, care is taken during sampling. The simulation runs are therefore sampled in a way that no data from one run can be taken into both the evaluation and training set. The sampled training set (90% of the samples) is then perturbed using a range of different noise levels using the approaches from Section 2.3. The whole evaluation methodology is illustrated in Figure 1, showing how the explanations are scaled and used throughout the process. The blue part of the graph illustrates existing work and systems, while the green part shows the novel scaling and scoring scheme.

#### 3.1.1. BLACK-BOX MODELS

The regressor ML models $f(\mathbf{x}_i)$ used for this evaluation are a linear regression model, a neural network with three layers of 32 neurons with Rectified Linear Unit (ReLU) activation modeled in PyTorch (Paszke et al., 2019), and the XGBoost regressor (Chen & Guestrin, 2016). Explanations $\mathbf{w}_i = \Phi(f, \mathbf{x}_i)$ are generated using the XAI methods discussed in the next section. The necessary gradients are calculated using either the parameters directly from linear regression, automatic differentiation for the neural network, or finite differences for the XGBoost tree. These local explanations are only generated on the unperturbed and complete evaluation set. This allows us to test how well and accurately ML models can learn feature importances even in noisy settings.

#### 3.1.2. EXPLANATION METHODS

The introduced measure $s$ and the three ML models $f(\mathbf{x})$ are then used in five different XAI methods. An overview of the general XAI landscape is provided in Section A.1, showing the different types of XAI methods and why we chose local explanations. The selected local XAI methods can generally be categorized in either *Effect-based Methods* (EM) and *Additive Methods* (AM). The first XAI method describes the effects the input shift has on the output; the latter how much

the feature contributes to the output - usually in a sparse form. The following XAI methods are used (theoretical details in Section A.2):

- The gradient baseline simply takes the gradient of the input of the ML model - a very simple approach (EM).
- Improving upon that, the SG method averages the gradient over $k = 10$ neighbors sampled using k-Nearest Neighbors (kNN) (Yeh et al., 2019) (EM).
- Next, the ALE-kNN (Apley & Zhu, 2019) uses a simple Gaussian conditional distribution with a fixed $\sigma^2 = 0.2$ scaled by the feature range, a resolution of $n_{samples} = 50$ bins over the feature range and $k = 10$ samples for the kNN selection (EM).
- LIME (Ribeiro et al., 2016) simply uses default values, with no modifications (AM).
- The SHAP (Lundberg & Lee, 2017) methods are used with default parameters. Different SHAP methods, however, are used, depending on the ML model - Linear SHAP for the linear regression, Tree SHAP for XGBoost, and Gradient SHAP for the neural network (AM).

## 4. Results & Discussion

This section presents the evaluation results, beginning with an overview of how noise affects the different XAI approaches, in combination with different ML models. Sanity checks with just noise are also performed, providing a lower empirical bar for $s$ of the evaluated systems.

### 4.1. Robustness Results

The core results of this paper from both datasets are shown in Figure 2 and 3, where we show the effect of progressively applying more noise to the training data. Various black-box models are then fitted to this perturbed training set and XAI methods are evaluated using the scoring method proposed in Section 3.

The first graphs of either dataset, Figures 2a and 3a, show the $R^2$ score, a metric to evaluate regression problems. The score can achieve a maximum of 1 if the predictions are perfect, 0 if they predict the mean and arbitrarily negative if the prediction is worse than the mean (Chicco et al., 2021). These scores show that the increasing noise has, as expected, adverse effects on the performance of the evaluation set. Furthermore, the linear regression model is quite robust in the regime of strong noise for the EAF, while it fails to capture any of the relations of the toy dataset. The next graphs of the toy dataset, Figures 2b through 2d show that the effect-based XAI methods are strongly dependent on the performance of the trained ML models. Linear regression fails to capture most relations while XGBoost and the neural network work quite well, especially in regimes of low noise using a robust

explainer like ALE-kNN. The additive methods, shown in Figures 2e and 2f, are not able to capture the sensitivity present in the ground truth $\mathbf{w}^*$ even without noise.

Similarly, the EAF results show that the explainer performance recorded is coupled with the ML model performance, as the effect-based methods fall with increasing noise. This rising noise plays, again, a key role in the lowering of the scores of the XGBoost and neural networks for the more robust XAI methods, SG and ALE-kNN, as seen in Figures 3c and 3d. These two and the raw gradient of Figure 3b show two further, interesting findings: first, when the linear regression performance is good, the explainer score of the linear regressssor is quite constant over the range of noise levels. Second, most ML models needed some initial noise on this dataset to start modeling the relations well, which is especially pronounced in Figure 3d, where all ML models rise slightly when adding a bit of noise. This could be due to a regularizing effect of the noise on the gradients, effectively creating more truthful averages of gradients when adding noise. The additive methods are again not as effective in explaining the ground truth sensitivity as evident in Figures 3e and 3f

### 4.2. Random baseline: Empirical lower bounds for Results

We additionally empirically calculate the random lower bounds using noisy data to make sure that the evaluation measures worked. This check was performed by training the ML model on the correct, slightly perturbed data using a noise level of $l = 0.05$. The ML model was then given random evaluation data, on which the scores for the interpreters were calculated. The evaluation dataset was set to $x_{ij} \sim \mathcal{N}(\mu_j, \sigma_j)$, where $\mu_j$ and $\sigma_j$ are the mean and standard deviation of the training dataset; the used dataset is the toy dataset. The results for this perturbation can be seen in Table 1, where all XAI methods approached a score of about 0.5 to 0.6, indicating, that this is indeed similar to the worst results observed above.

In the next experiment, we trained the ML model on completely noisy data, again with $x_{ij} \sim \mathcal{N}(\mu_j, \sigma_j)$, but scored on the correct evaluation set. This produced, expectedly, even worse performance metrics. This indicates that the random baseline for this scoring method is around $s = 0.55$.

### 4.3. Discussion

This investigation of the robustness of XAI methods showed that these approaches are influenced by the noise and the predictive performance. This is especially true for gradient-based XAI approaches. The SG as well as the cohort-based ALE-kNN works well. The properties of the custom cohort approach foster the correctness of the interpretations, as there is an uplift in performance, especially in combination

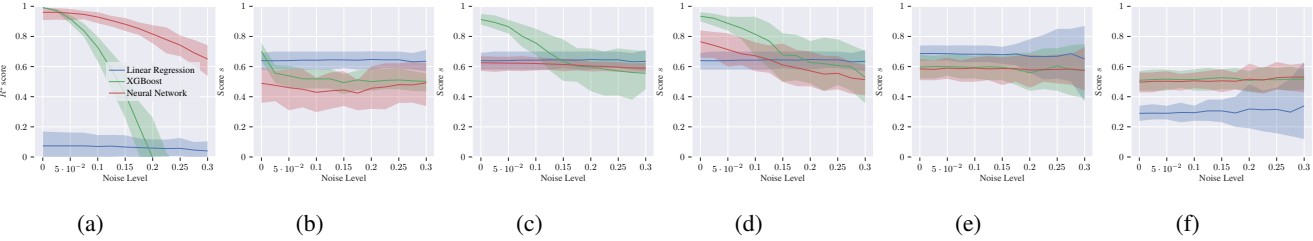

Figure 2: Score $s$ on toy data with varying levels of noise on the different combinations of explainers and ML models. The shaded area is the 90th and 10th percentile over 50 experiments with random sampling. (a) $R^2$ score, (b) Gradient score $s$, (c) SG score $s$, (d) ALE-kNN score $s$, (e) LIME score $s$, and (f) SHAP score $s$.

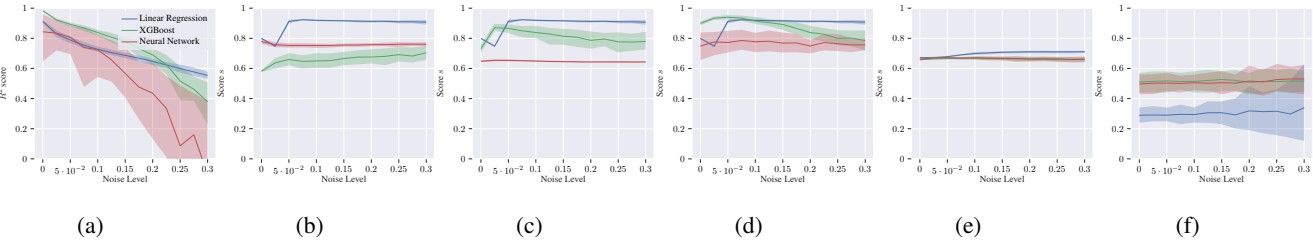

Figure 3: Score $s$ on EAF data with varying levels of noise on the different combinations of explainers and ML models. The shaded area is, again, the 90th and 10th percentile over 50 experiments with random sampling. (a) $R^2$ score, (b) Gradient score $s$, (c) SG score $s$, (d) ALE-kNN score $s$, (e) LIME score $s$, and (f) SHAP score $s$,

with the tree-based XGBoost system. Both LIME and SHAP are, however, in the category of additive methods. This makes them unsuitable for interpretations that call for scores reflecting the input sensitivity on the output, i.e. reflecting the feature *effect* of the output.

The low performance of the additive feature importance scores can also be partially attributed to the metric preferring the effect-based scoring methods, as the related work by Yeh et al. (2019) notes. Notwithstanding, we showed that effect-based scoring methods are highly dependent on the performance of the ML model to accurately reflect the ground truth importance scores $w_{ij}^*$ and that a single gradient of the ML model is often not enough to estimate them correctly.

## 5. Conclusion & Future Work

We showed how different XAI methods are affected based on the predictive performance of the ML models. The focus of this work was on model-agnostic post-hoc explanations for local data samples, as these could be evaluated using numeric, data-driven approaches. Of these XAI methods, SHAP, LIME, SG, and a local version of ALE were chosen. These were evaluated using a novel evaluation process focused on scaling the feature importance scores to a similar magnitude within one sample, then normalizing them to the same range as the ground truth effects, and, finally, calculating the distance to the ground truth effects reference. This ground truth was generated using a chemical simulation of

an EAF model, providing the necessary ground truth sensitivity for comparison and evaluation. This data-generating distribution was chosen based on the maturity of the EAF models for these real-world processes as well as current interest in the technology due to its promise of cleaner steel production. Additionally, a toy example was initially used to test the approaches on a limited and known nonlinear dataset.

Noise analysis over a range of perturbances of the initial dataset was performed using this evaluation methodology. The resulting analysis lends the conclusion that XGBoost in combination with smooth gradient-based XAI methods can approximate both the target values as well as the ground truth interpretations very well, even in noisy environments. LIME and SHAP, however, were not as successful in correctly finding the ground truth feature importance scores, probably due to their differing approaches to the interpretation of the feature importance scores. Some of these XAI methods and ML models, furthermore, showed a higher variance, indicating that they varied between sampling runs and were affected by the high noise. Sanity checks on the validity of the evaluation process were carried out using noise, first, as evaluation data, and then as training data. Both tests showed that the scores tend to be around $s = 0.55$ since the ML models cannot learn the importance at all.

Table 1: Results of sanity check with Gaussian noise as evaluation inputs on the toy dataset ($\pm$ one standard deviation).

| MODEL NAME | $R^2$ | GRAD | SG | ALE KNN | LIME | SHAP |
|---|---|---|---|---|---|---|
| LINEAR REGRESSION | -0.11±0.08 | 0.62±0.05 | 0.62±0.05 | 0.62±0.05 | 0.50±0.05 | 0.50±0.04 |
| NEURAL NETWORK | -1.05±0.33 | 0.48±0.06 | 0.56±0.02 | 0.53±0.06 | 0.53±0.05 | 0.50±0.05 |
| XGBOOST | -0.93±0.29 | 0.50±0.06 | 0.55±0.05 | 0.56±0.05 | 0.52±0.05 | 0.51±0.06 |

Table 2: Results of sanity check using Gaussian noise as training inputs on the toy dataset ($\pm$ one standard deviation).

| MODEL NAME | $R^2$ | GRAD | SG | ALE KNN | LIME | SHAP |
|---|---|---|---|---|---|---|
| LINEAR REGRESSION | -0.02±0.03 | 0.49±0.13 | 0.49±0.13 | 0.49±0.13 | 0.52±0.22 | 0.49±0.22 |
| NEURAL NETWORK | -0.03±0.08 | 0.48±0.11 | 0.53±0.01 | 0.47±0.09 | 0.51±0.16 | 0.51±0.15 |
| XGBOOST | -0.52±0.20 | 0.50±0.05 | 0.50±0.12 | 0.48±0.13 | 0.54±0.14 | 0.49±0.08 |

## 5.1. Future Work

There is an apparent need to quantify the uncertainty of the XAI methods, as this high variability of the feature importance scores of ML models with high uncertainty in noisy environments distorted the feature interpretation significantly. There are already works investigating such uncertainties for XAI (Löfström et al., 2024; Zhao et al., 2021; Slack et al., 2021), however, none of these address the effect-based approaches where the feature importance score reflects the change of the output with respect to the input.

The empirical evaluation of feature importance scores, especially from LIME and SHAP could also be further investigated by the comparison of different metrics on a ground truth dataset. Further improvements on the measures of infidelity and sensitivity (Yeh et al., 2019), combined with the consideration of a known ground truth feature importance and deeper analysis of noise could also lead to further understanding of robustness and failure cases of XAI.

## Acknowledgements

The financial support by the Austrian Federal Ministry of Labour and Economy, the National Foundation for Research, Technology and Development and the Christian Doppler Research Association is gratefully acknowledged. We furthermore thankfully acknowledge the financial support of the project by voestalpine Stahl GmbH. The authors gratefully acknowledge the funding support of K1-MET GmbH, whose research program is supported by COMET (Competence Center for Excellent Technologies), the Austrian program for competence centers. COMET is funded by the Austrian ministries BMK and BMDW, the Federal States of Upper Austria, Tyrol, and Styria, and the Styrian Business Promotion Agency (SFG).

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
