# A. Background & Related Work

## A.1. Interpretable and Explainable ML

Before diving into the XAI methods themselves, the notion of interpretability and explainability of ML models has to be examined first. These terms span a wide field of approaches, with the common goal of enabling the end users or creators of a system to foster their understanding on *why*, and sometimes *how* ML models arrive at certain conclusions. However, the presentation of these explanations can vary widely, from visual explainers to mathematical formulas. This paper will focus on the latter, as we will take advantage of that they are easier to compare and evaluate numerically in a data-driven environment. Most methods can, despite this variety in modalities, generally be categorized into either *interpretable systems* or so-called *post-hoc* methods (Murdoch et al., 2019; Doshi-Velez & Kim, 2017).

### A.1.1. INTERPRETABLE SYSTEMS

ML models can achieve inherent interpretability through different modalities, which, however, share the constraint of understandable processes that can be comprehended by a single person through a reasonable timeframe and effort. This limitation, coined by Murdoch et al. (2019) as "simulatability", greatly inhibits both the choice of system and the function space of the considered models. This leads, depending on the reduction in expressive power, to a loss of predictive performance and has to be considered before choosing such a model. The (numerical) evaluation of these systems is furthermore hindered by the wide-ranging type of approaches and often depends on human evaluation of these systems (Murdoch et al., 2019).

One of the two common ways to increase the range of available systems again can be the introduction of sparsity, where there is an initial, more complex function and feature space that can later be reduced by fitting the model to the data and reducing the set of expressions needed for decisionmaking. Similarly, the model could be broken up into smaller modules, where each is very well understood on its own - a popular choice for these modularized, linear components would be ensemble-based approaches with only a few models (Murdoch et al., 2019).

### A.1.2. POST-HOC INTERPRETABILITY

These interpretable approaches, however, do not foster the understanding of the model if there are no alternatives to the current model due to performance limitations or other constraining factors limiting choice. In these cases, post-hoc approaches can offer interpretability for black-box models that would be too complicated for humans to comprehend easily. These can be further categorized into global (dataset-level) and local (instance-level) approaches to interpretability. Global explanations are of interest for an overview of how the model considers certain features in general or for statistical analysis. If the interest lies in specific examples, or even failure cases such as adversarial examples, where a prediction can be maliciously flipped, a local explanation is often the needed approach (Murdoch et al., 2019). Furthermore, one can also group or cluster features and analyze these cohorts with global methods, achieving a similar overview-level interpretation on a smaller neighborhood (Munn & Pitman, 2022). All levels utilize similar techniques to arrive at solutions explaining the decision, with the core concept being called *feature scores*. These methods measure how influential a certain feature is, either on the local or global level, and assign it a score based on the perturbances it might achieve. These methods such as *partial dependence plots (PDPs)* and *ALE* (Apley & Zhu, 2019) (in the 1D case), or their local counterpart, *SG*, describe how varying a single feature affects the output, with the assumption that features of interest are not correlated. We will call these methods *effect-based* methods throughout this paper. *Feature interactions*, on the other hand, analyze exactly these contributions: how two or more features influence the output in conjunction (Naser, 2021). Other methods, such as *LIME* utilize simple, local representations of models as surrogates to better explain the behavior of models in certain neighborhoods (Ribeiro et al., 2016; Munn & Pitman, 2022). Both LIME and SHAP, another explanation methodology, form a group of systems we will call *feature importance scores*, where each feature is assigned a certain contribution to the output.

## A.2. Chosen Approaches

Of these methods, the local and cohort-based post-hoc approaches for feature importance are of interest for the evaluation task at hand, as these allow for comparison to known feature influences from the simulated data-generating process. This is why we chose ALE on a small cohort chosen using kNN (ALE-kNN), SG, LIME and SHAP.

### A.2.1. ALE-κNN AND SG

The intuition behind ALE (Apley & Zhu, 2019) is to simply accumulate the local effects (derivatives) of a function (model) weighted by a certain prior. This prior is the main difference to PDPs, as these do not consider the conditional distribution $p_{2|1}(x_2|x_1)$ of the sample, just the marginal over the selected variable. This results in a very intuitive and straightforward formulation of the ALE effect $f_{1,ALE}(x_1)$ on $x_1$ for a two-dimensional function ($d = 2$, $f(x_1, x_2)$) as

$$f_{1,ALE}(x_1) = \int_{x_{min,1}}^{x_1} \int p_{2|1}(x_2|z_1)f^1(z_1, x_2)dx_2dz_1 - \text{constant} \tag{7}$$

where $f^1(x_1, x_2)$ is the first derivative with respect to the variable of interest ($\frac{\partial f(x_1, x_2)}{\partial x_1}$). $x_{min,1}$ is chosen such that it just touches the lower support of the conditional distribution, approximating the integral over the whole distribution. The first integration variable $x_2$ integrates over all samples of the second, unrelated feature (or more in real-world examples), while $z_1$ integrates the effect over the conditional distribution. The resulting uncentered ALE is retrieved when ignoring the constant, which can be centered by subtracting the average effect. This method is usually applied on a dataset level and generates visual plots, showing the influence of variables over the whole range of the input (Apley & Zhu, 2019).

However, to apply this method to a local data sample, the $k$ nearest neighbors of the sample are chosen, and the resulting local cohort (Munn & Pitman, 2022) is used as the basis for the uncentered ALE of this sample. This local cohort version of the ALE is very closely related to the SG method, where the gradients or, more general, the explanations are averaged over a small neighborhood (Yeh et al., 2019). The version using ALEs has, however, the advantage of the introduction of the conditional distribution, effectively weighting the effect locally.

### A.2.2. LIME

LIME (Ribeiro et al., 2016), offers feature importance scores through local surrogate models, learning a sparse representation of the neighborhood. This approach combines almost all concepts outlined in the overview: simulatability, sparsity, local cohorts, and surrogate models. The resulting method returns feature importance scores. The algorithm itself consists of three major steps: sample selection, feature importance calculation, and a final selection process called "submodular pick". For local explanations, only the first two steps are necessary.

Sample selection searches a training dataset for similar datapoints using a configurable distance metric, and evaluates the model for each. This small set is then used for fitting the surrogate model to the local neighborhood. A linear regressor is usually used as a local surrogate. The weights of the resulting model are then directly utilized as the feature importance scores (Ribeiro et al., 2016).

### A.2.3. SHAP

The SHAP (Lundberg & Lee, 2017) framework unifies multiple methods for calculating SHAP values, a measure of feature importance. They are based on Shapley values, the only set of values satisfying the three desirable properties for additive feature attribution. This scoring method can be described as a simple linear model of the form

$$g(z') = \phi_0 + \sum_{i=i}^{M} \phi_i z_i', \tag{8}$$

where the model is similar to the function at $g(z') \approx f(h_x(z'))$ and $z' \in \{0, 1\}^M$ being a binary representation of $x'$, a *simplified input*, using the mapping function $h_x(z') = x$. The parameters $\phi_i$ describe the actual feature attribution.

While LIME follows the same structure, it does not necessarily fulfill the needed theoretical properties. Lundberg & Lee (2017) introduce, based on this notion, three properties. These properties are

1. Local accuracy: the linear model matches the complex model in the local point;
2. Missingness: the instances with $x_i' = 0$ have no impact, so $\phi_i = 0$; and
3. Consistency: a change of the model output w.r.t. one feature resulting in an output increase should change the attribution $\phi_i$ only positively.

The exact computation of Shapley values is quite challenging, as it is NP-hard - there are, however, heuristics and simplifications for existing models such as Kernel SHAP, which is model agnostic. Other methods such as Deep SHAP or

Tree SHAP utilize certain model-specific aspects to speed up the computation and provide more exact estimates. All of these approaches are united in the SHAP package (Lundberg & Lee, 2017).

### A.3. Related Work

Most of the mentioned methods prove their value either using human evaluation (Ribeiro et al., 2016; Lundberg & Lee, 2017; Ribeiro et al., 2018), with the use of toy examples (Ribeiro et al., 2016; Apley & Zhu, 2019; Shrikumar et al., 2017; Ribeiro et al., 2018) or improvements in theoretical properties (Lundberg & Lee, 2017). None of these works consider robustness to perturbances, unstable outputs, or other chaotic behavior. Some introductions of newer methods, however, are aware of these problems and decide to forego human evaluation in favor of data-driven analysis and robustness studies (Sun et al., 2020). Further works augment this missing robustness consideration in XAI, studying these problems in more depth (Alvarez-Melis & Jaakkola, 2018; Yeh et al., 2019; Bhatt et al., 2020; Huang et al., 2023; Crabbé & van der Schaar, 2023). Alvarez-Melis & Jaakkola (2018) and Yeh et al. (2019) tackle these problems through the maximal local sensitivity, where they search for the maximal change of the explanation to the instance within a neighborhood. This method asses the stability of local explanations and can thus be seen as a measurement for robustness, especially on noisy data. Other methods try to reduce the sensitivity through aggregation by combining different metrics into a more resilient measure (Bhatt et al., 2020). Yeh et al. (2019) introduces, besides the sensitivity of the explanation, the infidelity of explainable approaches, where they measure the squared difference from a projected explanation ($\mathbf{I}^\mathrm{T}\Phi(f,\mathbf{x})$) to the difference of the output when subtracting the projection from the input ($f(\mathbf{x}) - f(\mathbf{x} - \mathbf{I})$) using

$$\mathrm{INFD}(\Phi, f, \mathbf{x}) = \mathbb{E}_{\mathbf{I}\sim\mu_\mathbf{I}} \left[ \left( \mathbf{I}^\mathrm{T}\Phi(f,\mathbf{x}) - (f(\mathbf{x}) - f(\mathbf{x} - \mathbf{I})) \right)^2 \right] \tag{9}$$

where $\mathbf{I} \in \mathbb{R}^d$ representing the significant perturbances around $\mathbf{x} \in \mathbb{R}^d$ as a random variable, $\mu_\mathbf{I}$ the distribution of these perturbances, $f(\mathbf{x})$ the learned model function and $\Phi(f,\mathbf{x})$ the corresponding explanation. While this measure is more data-driven and universal, it still suffers from only using an approximation of the ideal explanation. Furthermore, no studies regarding the robustness with regard to added noise or known noise levels have been performed.