# OpenReview forum: "Robustness of Explainable Artificial Intelligence in Industrial Process Modelling"
_ICML.cc/2024/Workshop/ML4LMS — ML4LMS Poster_

### Official Review · Reviewer_ajet · 2024-06-09
**Review of Submission 3 - valuable discussion**

**Rating:** 7
**Confidence:** 5

**Review:**

The authors contribute to the field of XAI by proposing a new evaluation methodology and demonstrating its application in an industrial process modeling. The robustness analysis provides valuable insights into the effectiveness of different XAI methods for different ML models under varying noisy conditions.

I find such discussion very valuable because without reproducible, meaningful, and robust interpretation, ML models won't necessarily help us understand how or why they achieved accurate prediction or classifications. The introduction of a novel evaluation methodology for XAI methods based on ground truth sensitivity is also helpful.

Overall, the work could start some interesting discussions at the workshop.

A few suggestions would be on testing more ML models and definitely on more real-world datasets. XGBoost+gradient based XAI methods working well isn't super surprising, as they have a good record of being robust.

---

### Official Review · Reviewer_y5fJ · 2024-06-11
**Interesting benchmark of *post hoc* explainability methods in presence of noise.**

**Rating:** 6
**Confidence:** 3

**Review:**

**Summary Of Contributions:**

The paper presents the use of different standard *post-hoc* explainability methods for a machine learning method for a toy system and an industrially important system (Electric Arc Furnance). The authors try to draw conclusions from the relationship between the explainability and correct process description.

**Main Review:**

*Strengths*

1.	Different methods of explainability are tested.
2.	The impact of noise on the performance of explainability methods is evaluated.
3.	The normalization process seems interesting for further applications.

*Weaknesses*

1.	I am unclear about the justification for using the XAI methodologies in both problems. The first is a completely stationary problem, while the second seems to be time-dependent.
2.	The predicted quantity of the EAF process is not clear. As a result, it is also unclear how the model was trained and generated.
3.	I found it very interesting that the linear regression model is not sensitive to noise addition. Do the authors consider this a consequence of the addition of white noise?
4.	Is the number of features considered for the different explainers’ constant?
5.	Can the authors classify which features are more important for the different explainability methods? Is there any feature that is always more important than others?
6.	In view of industrial applications, do the authors evaluate how well their model performed when compared with experimental results?

**Summary of the Review:**

The paper presents how different explainability methods react under the influence of noise in the training data for a toy system and an industrial system. The results are interesting and demonstrate the role of noise in the interpretation. However, it is missing a sorting of the relevance of features and whether those can be related to specific parts of the model.

---

### Official Review · Reviewer_TZgR · 2024-06-12
**Analysis of explainable AI in Industrial Process**

**Rating:** 6
**Confidence:** 3

**Review:**

Just submitting the score for now, I will aggregate my comments on the paper and edit the response in the next couple of days